# VR Realism Scale—Revalidation of contemporary VR headsets on a Polish sample

Natalia Lipp[1]*, Radosław Sterna[2,3], Natalia Dużmańska-Misiarczyk[2,4], Agnieszka Strojny[1,4], Sandra Poeschl-Guenther[5], Paweł Strojny[1,4]

1 Institute of Applied Psychology, Faculty of Management and Social Communication, Jagiellonian University, Krakow, Poland, 2 Doctoral School in the Social Sciences, Jagiellonian University, Krakow, Poland, 3 Emotion and Perception Lab, Institute of Psychology, Jagiellonian University, Krakow, Poland, 4 R&D Unit, Nano Games sp. z o.o., Kraków, Poland, 5 Research Group for Media Psychology and Media Design, Institute for Media and Communication Science, Technische Universität Ilmenau, Ilmenau, Germany

☯ These authors contributed equally to this work.
* natalia.lipp@doctoral.uj.edu.pl

**Data Availability Statement:** All relevant data are within the paper and its S1 Appendix, S1 Dataset, S1 File files.

## Abstract

This paper presents validation of the VR Simulation Realism Scale on a Polish sample. The scale enables a self-report measurement of perceived realism of a virtual environment in four main aspects of such realism–scene realism, audience behavior realism, audience appearance realism and sound realism. However, since the development of the original scale, the VR technology significantly changed. We aimed to respond to that change and revalidate the original measure in the contemporary setting. For the purpose of scale validation, data was gathered from six studies with 720 participants in total. Five experiments and one online survey were conducted to examine psychometric properties of the scale in accordance with the Standards for Educational and Psychological Testing. Evidence based on internal structure, relations to other variables and test content was obtained. The factorial structure of the original scale was tested and confirmed. The connections between realism and immersion, presence, aesthetics were verified. A suppressed relationship between realism and positive affect was discovered. Moreover, it was confirmed that scale result is dependent on the quality of VR graphics. Results of the analyses provide the evidence that the VR Simulation Realism Scale is a well-established tool that might be used both in science and in VR development. However, further research needs to be done to increase external validity and predictive power of the scale.

## Introduction

### Immersive virtual environments

Virtual Reality (VR) is usually defined as technology (hardware) that uses various human-computer interfaces to create the impression of being in a virtual world [1]. Nowadays in the field of psychology, virtual reality is presented in the context of so-called Immersive Virtual

**Funding:** The work of RS was financed from the science budget (https://www.gov.pl/web/edukacja-i-nauka/diamentowy-grant) for the years 2019–2023 as a research project, 'The impact of virtually generated characters: human aspects of bots,' under the Diamond Grant program awarded by the Ministry of Science and Higher Education of Poland (project number: DI2018 015848). Studies A–E were supported by the Polish National Centre for Research and Development (https://www.gov.pl/web/ncbr) under the Widespread Disaster Simulator grant – research and preparation for implementation received by Nano Games sp. z o.o. (project number: POIR.01.01.01-00.0042/16 the Smart Growth Operational Programme, sub-measure 1.1.1. Industrial research and development work implemented by enterprises). Study F was supported by the Polish National Science Centre (https://www.ncn.gov.pl/) under the grant 'Wirtualna śmierć realne konsekwencje - efekty śmierci bohatera w grze wideo w świetle Teorii opanowania trwogi' [ang. Virtual death – real consequences. The effect of virtual agent's death according to the terror management theory] received by AS (project number: DEC-2018/02/X/HS6/01426). The proofreading of this article was funded by the Priority Research Area Digiworld under the program Excellence Initiative – Research University at the Jagiellonian University in Krakow. The open-access publication of this article was funded by the Faculty of Management and Social Communication at the Jagiellonian University in Krakow. Nano Games sp. z o.o. also provided support in the form of salaries for AS, PS, NDM, NL, and RS. The specific roles of these authors are articulated in the 'author contributions' section. The funders had no role in study design, data collection and analysis, decision to publish, or preparation of the manuscript.

**Competing interests:** The authors have read the journal's policy and have the following competing interests: AS, PS, NDM, NL, and RS were employees of Nano Games sp. z o.o. at the time when the study was designed and conducted. The studies were conducted using the VR Emergency Treatment Simulator (pol. Symulator Katastrof Zbiorowych) created by Nano Games. This does not alter our adherence to PLOS ONE policies on sharing data and materials. There are no other patents, products in development or marketed products associated with this research to declare.

Environments (IVE), which can be defined as synthetically produced sensory stimuli that surround the subject perceptually and give the perception that these environments and their components are not synthetic [2].

Immersive Virtual Environments (IVE) have been used for years as educational and training tools [3]. Originally, IVE were used for military or surgical training [4, 5] and were based mainly on algorithmic sequences related to motoric human-system interactions. These early applications can be considered traditional [6]; however, as technology has developed, researchers have begun to use IVE in other fields as well. Now, this technology can be used for, among others, decision-making practice, social skills training, or psychotherapy [7, 8]. The evolution of IVE applications progressed from manual motor-focused traditional simulators to more sophisticated psychologically focused tools. As technological capabilities continue to increase, modern IVE are becoming more similar to the physical world. However, the pursuit of extreme realism may not be justified as it seems that objective realism and its subjective perception (i.e., simulation realism [9]) do not always overlap. Something that is perceived as realistic does not have to be a perfect representation of reality in a virtual environment [10].

## Subjective measurement of simulation realism

The need for a self-report measure of simulation realism arose from the fact that, as mentioned above, objective realism (interaction and display fidelity [9]) and subjective perception of it (simulation realism [9]) may differ from each other, while the latter is important for task performance, skill transfer and feelings of presence in a virtual environment [11]. In order to respond to this need, a self-report measure was created: items from the Witmer-Singer Presence Questionnaire [10] were translated into German, and items describing the realism of a virtual audience's appearance and behavior were constructed and integrated into a 14-item questionnaire [11]. This questionnaire was tested on a sample of $N = 151$ participants in a CAVE application for addressing fear of public speaking.

Varimax orthogonal rotation was used for the factor analysis; four factors were discovered that explained a total of 69.37% of the variance [11]:

1. Scene Realism–fidelity of features such as shadows, lights, reflections, and colors (5 items);

2. Audience Behavior–fidelity of the gestures, postures and facial expressions of a virtual audience (4 items);

3. Audience Appearance–fidelity and realism of a virtual audience's characteristics, such as adequateness of clothing and the diversity and general authenticity of virtual humans (4 items);

4. Sound Realism–a measure of the perceived adequateness of volume (1 item).

The reliability of the original scale is satisfying. It has been used in several studies since it was created; however, some of the items were derived from earlier questionnaires created in the 1990s [10], so it is doubtful that they are still useful for assessing the simulation realism of modern virtual environments. Therefore, in light of growing interest in studying the nature of realism and the fact that CAVES and modern headsets differ greatly in terms of the characteristics of the stimuli they deliver [12], we decided to examine the psychometric properties of the scale in the context of state-of-the-art VR technology and a modern IVE.

The rationale for choosing this scale was also that its items are fairly universal in their formulation and thus can be used to assess different hardware and technology without making any adjustments or changes to the scale. By doing this, we create a solid baseline for interpretation of our findings on simulation realism and its impact on other psychological variables.

We think that validation of existing tools is of special importance in the VR research field. As shown in a recent paper by Fitrianie and associates [13], there is a trend in VR research to create new questionnaires continuously. This occurs even in situations in which there are already established tools in the literature, as was shown by Oh and colleagues [14] in the context of social presence, for which over 40 questionnaires have been developed to measure this construct. Instead of following this direction, we wanted to make use of and revalidate an existing questionnaire, thus promoting its usage in the field of VR research.

Moreover, this approach, in which we validate existing questionnaires instead of producing new ones, can result in standardization of the methods used and ultimately to increased comparability of studies' results.

As mentioned before, the original scale was validated on a CAVE system, but these are undeniably less popular nowadays than VR headsets. Therefore, one of the motivations for the study was to test the scale with a different device. Before proceeding to widespread usage of the VR realism scale, it is important to make sure that it performs sufficiently.

Furthermore, the validation of this questionnaire in Polish can be justified from the perspective of the dynamic growth of the Polish game industry. According to a recent report on this industry [15], there are 440 development studios in Poland, and 96% of games produced in Poland are exported. Moreover, in 2018 and 2019, 68 games for different virtual reality technologies were released. The game industry in Poland generated a revenue of €479,000,000 in 2020, and there were 16,000,000 gamers in Poland in 2020. As can be seen from these data, gaming is a big industry in Poland. Tools such as the VR realism scale could be used to validate different gaming and serious gaming tools.

Not only the game industry is growing in popularity in Poland: the scientific community in Poland has also started to show interest in conducting studies using various virtual environments [16–19].

## VR-related variables

The most common concepts which are brought to the discussion about the virtual experience are immersion, presence, co-presence, flow, and simulation realism. Although simulation realism is the variable of our interest, we present immersion and presence first as these are the basis of the virtual experience.

Immersion is defined as an attribute of a medium that allows the user to experience an integral and extensive illusion of reality [20]. In turn, presence is defined as a 'state of consciousness, the (psychological) sense of being in the virtual environment' [20 p605]. A high level of immersion is required to induce presence, but the influence of immersion on presence may be not direct. Simulation realism (defined as the extent to which virtual objects are perceived as real [11]) may be considered as a variable that mediates between immersion and presence. In other words, when immersive technology is perceived as real, then presence is induced [21]. In addition to these concepts, some researchers highlight the role of co-presence (the social aspect of the virtual experience [22]) and flow state. Flow is a well-known construct that describes the feeling of being fully involved in and enjoying an activity [23]. Additionally, it has been shown that flow is strongly related to the sense of presence and better performance of VR tasks [24]. These processes are the most frequently described concepts in the field of VR studies and are often listed as factors behind an effective IVE experience [25–27]; on the other hand, we still do not know how they work in an IVE and what conditions must be met for these psychological processes to be triggered.

It seems that all these factors might have a common origin: the perception of the virtual as real [21, 25, 28]. As stated before, the degree to which virtual stimuli are perceived as real by

the user is called simulation realism [11]. This definition applies mainly to visual realism (i.e., faithful replication of objects, [29]); however, in the course of research on this construct, different types of realism have been identified (realism of interaction, realism of behavior, etc. [25]). It is important to note that realism in this context relates to the experience of the environment and not to its objective characteristics. Bowman and McMahan [9] proposed the term *fidelity*, which consists of three parts: the system's output (*display fidelity*), the exactness of possible interactions (*interaction fidelity*), and the realism with which the physical world is rendered in the virtual environment (*realism of simulation*).

The influence of both display and interaction fidelity on the virtual experience has been verified in many studies [30–32]. In contrast, simulation realism is often not assessed as part of IVE evaluations due to difficulties with its measurement [11]. Perhaps the reason behind researchers' preference for testing display and interaction fidelity is the direct relation between them and the capabilities of technology. In contrast, simulation realism is not an objective measure and depends more on cognitive representations and perception of virtual models than on hardware or software. Nevertheless, attempts are being made to operationalize simulation realism and to test the extent of its impact on other variables as it represents a fundamental concept in users' reactions to a virtual environment.

## Relationships between simulation realism and other VR-related variables

Concerning the relationship between presence and simulation realism, the research is inconsistent. Some works point to the importance of pictorial realism [33], but others emphasize the importance of consistency between behavioral and pictorial realism. All elements of a virtual scene should have the same level of realism [34].

A similar issue can be observed in regard to co-presence and its relation to the realism of virtual characters. Some studies show that realism of behavior is the most important factor in increasing copresence [35–37], while others show that some compatibility between appearance realism and behavioral realism is crucial [38, 39].

The definition of simulation realism implies that virtual stimuli are assessed in terms of how similar they are to the corresponding real objects. However, realism is also an art movement, which could imply that VR simulation realism is an aesthetic category. Aesthetics is a construct that describes a subjective pleasurable experience while engaging with stimuli. The connection between simulation realism and aesthetics may imply other consequences as aesthetics is known for its association with pleasure and evoking positive emotions [40, 41]. Therefore, it is possible that assessment of virtual stimuli as real may induce pleasure and positive affect.

## Aims of our study

The main aim of our study was to validate the VR Realism scale [11]. According to Messick [42], validity is 'an evaluative summary of both the evidence for the actual as well as the potential consequences of score interpretation and use' [42 p5]. As Kazi and Khalid [43] note, validation is a process which ensures that a tool measures what it was made to measure. Moreover, validation helps researchers collect better-quality data.

Although validity is a unitary concept, there are many ways to analyze and demonstrate it by referring to different aspects of it [44]. *Standards for Educational and Psychological Testing* [45] lists five sources of validity evidence: internal structure, relations to other variables, test content, consequences of testing, and response process. Our goal is to test the VR Realism scale according to these standards.

Firstly, we aimed to examine the psychometric properties of the VR Realism scale. For this purpose, we verified its internal structure by analyzing its factorial structure and internal consistency. For factor analysis, we chose a confirmatory approach based on the original structure of the scale. It should be noted here that the Sound Realism scale may appear controversial as it consists of only one item. We were aware that the use of a single item to measure a latent variable is questionable; however, we found that removing it before testing the Polish language version of the scale might be considered premature and could lead to difficulties in comparing models and interpreting results. Therefore, we decided to test the model exactly as it was originally created by the German team using a single item related to sound realism. We further address our findings and recommendations in the discussion.

Secondly, we aimed to explore the connections between realism and other variables that influence human-computer interaction. When validating a scale, it is crucial to show a pattern of external relationships to similar constructs. These relationships should be consistent with expectations based upon theoretical assumptions [46]. Thus, we decided to correlate the score on the scale with the main characteristics of the experience of being in an IVE: presence, co-presence and immersion. Based on previous studies, we expected that the highest positive correlation coefficient would be observed between simulation realism and immersion. As proposed by [9], the term 'fidelity', of which simulation realism is a component, should replace the term 'immersion', therefore these two constructs are strongly connected. Moreover, we expected that simulation realism would correlate strongly and positively with presence, especially the realness factor [33]. Further expectations concerned a high positive correlation between simulation realism (in particular, its social factors: audience behavior and audience appearance) and co-presence [2]. We also expected a moderate positive correlation between simulation realism and aesthetics. The beauty of realism may not be surprising in the light of research on aesthetic assessment because users tend to prefer well-known objects which are similar to their prototypes [47]. The role of familiarity in the assessment of aesthetics has been verified in many studies [44, 45, 48]. In these terms, aesthetic judgment may be evoked by simulation realism, which is defined as the extent to which a virtual environment is perceived by the user as a credible representation of the real world.

We also expect aesthetics to be a mediator between realism and positive affect because aesthetics is related to pleasure. However, to the best of our knowledge, no study has yet verified this assumption, although the relationship between positive emotions and presence is well documented in the literature [49]. It seems that the virtual experience is inherently positive; however, there is no explanation of why positive affect is evoked during a virtual session. We assume that aesthetic assessment of a simulation is related to an increase in positive emotions.

After verifying the relationship between simulation realism and similar constructs (immersion, presence, co-presence and aesthetics), we wanted to explore the boundaries of the realism construct by testing the discriminant validity of the VR realism scale. Psychological constructs tend to overlap [50–53], so our analyses aimed to reduce the definitional ambiguity of realism. To show the discriminant validity of the scale, we chose flow and satisfaction of players' needs [54]. Although flow may be considered to be a factor that affects presence [55], there is no evidence that simulation realism is involved in evoking flow. The same applies to players' needs satisfaction. Although satisfaction of players' needs is important for their well-being, engagement, and therefore for a full virtual experience [56], to the best of our knowledge there is no indication that simulation realism is an underlying mechanism. Therefore, we expect these variables to correlate poorly or not at all with simulation realism as these constructs are not related to simulation realism in terms of content.

Additionally, we believe that there is one more reason to verify the relations between simulation realism and other constructs that describe the experience of an IVE. As was mentioned

in the Introduction section, highly efficient simulators are those which induce high levels of immersion, presence and co-presence [57]. To induce these, it might be necessary to create sufficiently realistic stimuli, but the exact relationship between all of the discussed variables is still unknown. The data obtained in our research might help in the creation of effective simulations.

Thirdly, our aim is to verify test content, which is 'the degree to which the content of a test is congruent with testing purposes' [58 p101]. To support this type of evidence, firstly we need to determine the main purpose of using the VR Realism Scale. We believe that this scale's score can be used as an assessment of a simulation and as a predictor of users' virtual experience. To prove this, we examined the VR Realism Scale to determine whether it is sensitive to small changes in graphics quality, and we explored the relationship between realism and positive affect.

Our work might be a step towards a better understanding of the virtual experience. On one hand, we aimed to provide researchers and developers with confirmation of the structure, usefulness and sensitivity of a well-tested tool that is used to measure one of the crucial aspects of the IVE experience. On the other hand, we aimed to explore the key psychological characteristics that lead to an effective simulation and its relation to realism.

## Materials and methods

### Tested samples

The total pool of collected data consists of seven studies (A–F), conducted between the years 2017 and 2019. Five of them (B–F) were experimental studies and one (A) was an online survey. The participants in study A were video game players; the participants in studies B, D, and E were cadets from the College of the State Fire Service and active firefighters from firefighting units in Cracow. University students participated in studies C and F. The number of participants and their basic demographic information are presented in Table 1. Information about the analyses performed on subsamples of data is also given in this table.

### Procedures

Study A was an online survey in which the participants responded to several questionnaires concerning a video game they had recently played. The participants of the study provided informed active consent before the study protocol. Studies B–F were experiments that were conducted in a three-dimensional virtual environment. There were two groups in study B: an experimental one (where the task was to conduct a rescue operation on virtual victims in a

**Table 1. Summary of data used for the purposes of scale validation and demographic information about the participants.**

| Study | $N$ | Women | Men | Age $M(SD)$ | Min. age | Max. age | Date of study | Performed analyses |
|---|---|---|---|---|---|---|---|---|
| A | 245 | 59 | 186 | 24.1(4.47) | 18 | 40 | 2017/07–2017/08 | correlation analysis |
| B | 60 | 1 | 59 | 21.58(1.45) | 19 | 24 | 2017/11–2017/12 | Item analysis CFA with measurement invariance |
| C | 60 | 33 | 27 | 22.32(1.63) | 20 | 27 | 2018/01 | Item analysis CFA with measurement invariance |
| D | 121 | 2 | 119 | 24.4(5.63) | 19 | 42 | 2018/02–2018/03 | Item analysis CFA with measurement invariance correlation analysis $t$ test |
| E | 111 | 2 | 109 | 23.66(5.02) | 19 | 42 | 2018/04–2018/05 | $t$ test mediation analysis |
| F | 120 | 60 | 60 | 21.13(2.05) | 18 | 29 | 2019/10 | Item analysis CFA with measurement invariance |
| Total | 720 (610 unique) | 157 (155 unique) | 563 (455 unique) | – | 18 | 42 | – | |

virtual environment) and a control one (where the participants were asked to explore the same virtual environment freely but with no victims present). Study C was designed to test the social facilitation effect, so there were virtual bystanders at the scene in two of the four groups. The conditions also differed in terms of the difficulty level of the task (moving bollards from one side of the street to another). Studies D and E were part of a longitudinal project; the virtual environment used in these studies differed only in terms of the quality of the sound and graphics. The task was the same as in the experimental group of study B. Study F was designed to test mortality salience in a virtual context. The task in this study was to find out what had previously happened in the virtual environment and to secure the scene. There were four experimental conditions in a 2x2 factorial design: the death of a virtual agent vs. all agents alive x task described as a fun game vs. as a simulator for critical infrastructure operators. All the described studies were accepted by the Ethical Committee of Jagiellonian University at the Institute of Applied Psychology. As the number of studies utilized for the present analyses is large, a detailed description of all the procedures is provided in the S1 File. The summary of all procedures is presented in Table 2.

**Table 2. Summary of study procedures.**

| Study | Type | Virtual Environment | Aim of the study | Procedure |
|---|---|---|---|---|
| A | Online survey–correlational study. | No actual VE (study conducted using Survey Monkey). | Identification of variables describing a full virtual experience. | Participants were asked to recall the last game they had played and to complete several questionnaires. |
| B | Experiment with physiological and questionnaire measurement. | VR simulator for rescue services with a scene presenting a car crash with multiple victims. | Assessment of level of arousal, workload, and emotions during a rescue action in VR. | Participants were randomly assigned to one of two conditions (experimental or control group) and asked to perform a given task in a VR simulator, during which physiological measurement was conducted. Subsequently, participants completed a set of questionnaires. |
| C | Experiment with physiological, behavioral and questionnaire measurement. | VR simulator with a small town scene. | Examination of the social facilitation effect in a virtual context. | 2x2 (task difficulty x presence of virtual agents) between-subject design was used. Participants were asked to perform a previously practiced task in a VR simulator. The task was to move virtual objects from one side of the virtual street to the other. Completion time and EDA were measured. |
| | | | | Subsequently, participants completed a set of questionnaires. |
| D | Experiment with physiological and questionnaire measurement. First iteration of the longitudinal study. | VR simulator for rescue services with a scene presenting a car crash. | Increasing the level of firefighters' engagement during a rescue operation. | Between-subject design with four conditions. There were three experimental groups with different distractors (e.g., virtual bystanders, a dog) and one control group. The participants had to perform a rescue operation during which EDA, ICG and ECG were measured. Subsequently, participants completed a set of questionnaires. |
| E | Experiment with physiological and questionnaire measurement. Second iteration of the longitudinal study. | VR simulator for rescue services with a scene presenting a car crash. | Increasing the level of simulation realism. | The procedure and measures were identical to study D but with several minor changes in the distractors. Several changes were also made to the virtual scenario. |
| F | Experiment with questionnaire measurement. | VR simulator with a small town scene. | Examination of mortality salience effects in a virtual context. | 2x2 (death of virtual agent x serious or fun context) between-subject design was used. The task of the participants was to find out what had happened in the virtual environment and to secure the scene of the event. After task completion, participants were asked to fill out several questionnaires. |

## Translation process

The original instrument consists of 14 items and was validated in German. After obtaining the authors' written consent, the original items were translated into Polish by a sworn German translator. In the next step, we conducted a pilot study on five judges who are competent in the field of psychology in order to evaluate the instructions, items, and response format clarity. During this process we identified minor language imperfections in the Polish translation and, in cooperation with the translator, we decided to reformulate the final wording of these items. The final Polish version was consulted with a linguist and back-translated to German by a bilingual German resident born in Poland. Both German versions were compared to each other by the authors of the original scale, who stated that they are satisfactory. The final Polish version was established without further amendments. The Polish version of the scale may be found in S1 Appendix.

## Measures

The data came from six different studies which dealt with different research questions and hypotheses. Therefore, this section is limited to a description of the measures that will be used to assess the external validity of the VR Realism Scale. We chose these measures because in our opinion they suffice for full description of the VR experience. A description of all the measures used can be found in S1 File. For an overview of which methods were used in which study, see Table 3.

**The Polish adaptation of the Igroup Presence Questionnaire** (IPQ; [59]) by Strojny, Lipp, and Strojny (unpublished) was used to measure the sense of presence (in three dimensions: spatial presence, involvement, and realness). It consists of 13 items, and participants indicate their answers on a 5-point Likert scale. The psychometric evaluation of the Polish version revealed satisfying internal consistency coefficients (Cronbach's alpha for the three factors > .80).

**The Players' Needs Satisfaction Questionnaire** [54] is based on Self-Determination Theory. It measures the level at which three universal needs (competence, autonomy, and relatedness) are satisfied by playing a game; this questionnaire also measures presence/immersion

**Table 3. Self-report measures used in the reported studies–an overview.**

| Questionnaire | Study A | Study B | Study C | Study D | Study E | Study F |
|---|---|---|---|---|---|---|
| **Realism Scale** [11] | X | X | X | X | X | X |
| **Igroup Presence Scale** [59] | X | X | X | X | X | |
| **Player Needs' Satisfaction Questionnaire** [54] | X | | | | | |
| **Immersion Questionnaire** [60, 61] | X | | | | | |
| **The Flow State Scale-2** [62, 63] | X | | | | | |
| **Scale of Aesthetics** [40, 64] | X | | | X | X | |
| **Scale of Mood** [65] | | X | | | | |
| **Scale of Emotions** [65] | | X | | X | X | |
| **NASA Task Load Index** [66, 67] | | X | | X | X | |
| **The General Self-Efficacy Scale** [68, 69] | | X | | | | |
| **Stress Appraisal Questionnaire** [70] | | X | | X | X | |
| **Self-assessment Manikin** [71] | | X | | X | X | |
| **Co-presence Scale** [72] | | | X | X | X | |
| **Simulator Sickness Questionnaire** [73, 74] | | | | X | X | X |
| **Positive and Negative Affect Schedule** [75, 76] | | | | | | X |

and intuitive controls. In the absence of a properly validated Polish version of this questionnaire, we decided to assess the internal consistency of the translated version using Cronbach's alpha reliability coefficient. The obtained Cronbach's coefficients were as follows: .79 for competence need, .80 for autonomy need, .67 for relatedness need. The coefficients of the original scale were .63 for competence, .71 for autonomy, and .72 for relatedness.

**The Immersion Questionnaire** [60, 61] was used to measure the players' absorption in the virtual environment. It consists of 27 items. Factor analyses performed by its authors confirmed the one-factor structure of this scale. The authors of the Polish version obtained a high reliability coefficient (Cronbach's alpha = .94).

**The Flow State Scale-2** [62, 63] assesses the experience of flow during a game session. It contains 36 items measuring nine aspects of flow. For the current analysis, the flow variable was calculated as the average result of all items. According to the authors of the original scale, the reliability of each individual subscale (tested in two studies) ranges from .80 to .90 with a mean of .85, and from .80 to .92 with a mean of .87.

**The Scale of Aesthetics** [48, 64] is a 10-item scale that was used to evaluate perceived aesthetic aspects of graphics quality (in the classical and expressive dimensions). The classical dimension describes the order, clarity and familiarity of a design, while the expressive dimension describes its originality, richness, creativity and novelty. Participants answer the questions on a 7-point Likert scale. According to the authors of the original French scale, both subscales have a reliability of .86 [77]; the Polish version has very similar reliability (.86 and .87 for classical and expressive, respectively).

**The Polish translation of the Co-presence Scale** [72] was used to measure social aspects of VR, particularly engagement in relationships with virtual characters (in four dimensions: presenter's reactions to virtual agents, perceived virtual agents' reactions, the impression of interaction possibilities, and the (co-)presence of other people). The items are rated on a 5-point Likert scale. Due to the lack of a properly validated Polish version, we assessed the internal consistency of the Polish translation using reliability analysis. The obtained Cronbach's coefficient was high (alpha = .89). In the original German version, the reliability of the first three subscales is high (alpha > .80); for the "(co-)presence of other people" subscale it is slightly lower (alpha = .71).

**The Scale of Emotions** [65] was used to assess the intensity of six basic emotions: joy, love, fear, anger, guilt, and sadness. It consists of 24 items rated on a 5-point Likert Scale. The alpha coefficients of the scales are as follows: alpha = 0.81 for joy, alpha = 0.82 for love, alpha = 0.80 for fear, alpha = 0.85 for anger, alpha = 0.55 for guilt, and alpha = 0.86 for sadness.

**The Self-Assessment Manikin** [71] (SAM; Bradley and Lang 1994) is a pictorial questionnaire. It was used to assess emotional responses to stimuli in three fundamental dimensions: valence, arousal, and dominance.

## Data analysis strategy

Several statistical procedures were applied to the data. In this section, they are outlined and briefly described.

**Preliminary analyses.** Before the main analyses, several initial steps were applied. Firstly, since the used data comes from six different studies (A–F), the General Linear Model (GLM) was used to test how much of the total variance in the data is explained by the data source (a single study). We decided that if Eta-squared was lower than 0.03, the source of data would be omitted from the analyses.

**Internal consistency—Reliability analysis.** In the first step, we performed an item analysis to examine the indexes of discrimination. We chose the corrected item-total correlation,

which is defined as a correlation between one selected item score and the total scale score (excluding the selected item [78]). The interpretation of the corrected item-total correlation coefficients is as follows: r < .19 indicates that the item does not discriminate well; values between .20 and .30 indicate good discrimination; r > .40 indicates very good discrimination. It is suggested that items should be dropped if they correlate negatively with the total scale.

Reliability analysis was performed to evaluate the internal consistency of the scale. A scale is considered to be homogeneous if the Cronbach's alpha coefficient is higher than .70 [79]. However, .70 reliability may not be accurate enough and it was proposed that .80 (or higher) alpha coefficient should be used for satisfactory internal consistency [80]. Since Cronbach's alpha is sensitive to the number of items, it should be noted that one could expect this coefficient to be relatively weak when it is calculated for separate subscales.

**Internal structure of the scale—Factorial analysis.**   The factorial structure of the Polish version of the Realism Scale was compared to the original structure postulated by Poeschl and Doering [11]. Confirmatory factor analysis was conducted using R's lavaan package [81] and the diagonally weighted least squares estimation (DWLS) procedure.

It was decided that the evaluation of the model's fit would be based on fit indexes rather than on the Chi-square goodness of fit, which is well known to be sensitive to larger sample sizes [82]. As recommended [82], two incremental indexes (TLI, CFI) and two absolute indexes (SRMR, RMSEA) were chosen for evaluation of the model's fit. The proposed criteria for the chosen indices are as follows: TLI > .95, CFI > .95, RMSEA < .08, SRMR < .08 [82].

As our validation was performed on a different (Polish) sample using different technology (VR HMD) than the original scale, which was validated on a German sample using CAVE, it is important to test measurement invariance in order to ensure that the measured constructs mean the same across distinct groups. Measurement invariance may be conducted using multigroup confirmatory factor analysis with a series of models. Each subsequent model is more restrictive in terms of the number of parameters that are set to be equal across groups. In the first step, configural invariance is tested. Configural invariance is least restrictive as it allows all parameters to vary freely across groups. It provides evidence of the similarity of the tested model's structure. In the next step, metric invariance, which constrains factor loading to be the same across groups, is verified. Metric invariance indicates that participants of both groups understand the constructs in the same way. If the model holds, the factor loadings and item intercepts can be constrained to be qual (scalar invariance). Scalar invariance makes it possible to assess the mean difference of the latent variable across groups. Lastly, the residual invariance is tested; this is the most restrictive model as factor loadings, item intercepts, and items' residual variances are set to be equivalent across groups [83, 84].

Measurement invariance is evaluated by comparing subsequent pairs of models (i.e., configural vs metric, metric vs scalar, scalar vs residual). To assess measurement invariance, the following indexes are considered: chi$^2$, CFI, RMSEA, BIC, and AIC. The CFI and RMSEA interpretations are the same as in the case of confirmatory factor analysis. Non-invariance can be identified based on a decrease in goodness-of-fit indexes. Additionally, AIC and BIC refer to predictive accuracy and are measures of comparative fit. This means that the model with the lowest BIC and AIC predicts new data most accurately [84].

**Evidence based on relations to other variables—Correlation analysis.**   To obtain evidence based on relations to other variables, we chose correlation analysis. We decided to test variables that are both related and unrelated to the construct. Such an approach allows both convergent and discriminant evidence to be obtained [58]. According to the *Standards*, "relationships between the test scores and other measures intended to assess similar constructs provide convergent evidence, whereas relationships between test scores and measures purportedly of different constructs provide discriminant evidence" [85 p14].

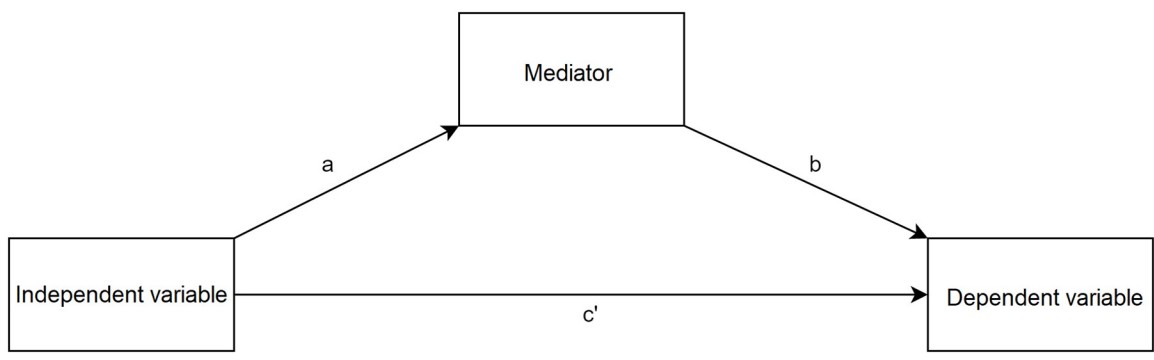

**Fig 1. Graphical representation of the mediation analysis [90].**

**Evidence based on relations to other variables—Mediation analysis.** We assumed that simulation realism may be related to aesthetics and therefore may evoke positive affect. Aesthetic assessment may be an underlying process that links realism and emotions. In order to explore this mechanism, mediation analysis was used. We chose mediation analysis because it allows the hypothesis about a third variable's influence to be tested, which may be crucial for understanding the mechanism by which an effect operates [86]. Mediation analysis tests the relationship between two variables (direct effect) but it also tests the relationship between three variables (indirect effect). Simple mediation analysis is based on three assumptions. Firstly, there must be a statistically significant relationship between the independent and dependent variables (path c' in Fig 1). Secondly, the influence of the independent variable on a mediator must be statistically significant (path a in Fig 1). Lastly, a mediator must significantly influence the dependent variable (path b in Fig 1 [87]). However, if there is a possibility that mediation analysis does not meet the first criterium, then it is called suppression analysis [88], which is mathematically equivalent to mediation. A suppressor is a third variable that increases the direct effect [89].

For the mediation analysis, the SPSS Process Macro [90] was used. This macro tests the mediation hypothesis with the use of a bootstrapping procedure. We computed unstandardized indirect effects for each of the 5,000 bootstrapped samples and the 95% confidence intervals (CI) by determining the indirect effects at the 2.5th and 97.5th percentiles. For each path, beta coefficient and confidence intervals are provided.

## Results

### Preliminary analyses

The GLM analysis revealed that less than 9% (eta-squared = .086) of the total variance stems from the source of the data. Thus, we included the data source (a study) as a covariate in the CFA analysis. All data used may be found in Supporting Information (S1 Dataset).

### Internal consistency—Item analysis and reliability analysis

The discrimination indexes were examined by item analysis. To perform item analysis, corrected item-total correlation was chosen. The analysis was performed twice: first, to obtain correlation coefficients for the total scale (including all 14 items); second, to obtain correlation coefficients for the three subscales.

The results are presented in Table 4. Except for item 14 (i.e., sound realism; rjx = .07), the coefficients indicate good discrimination ranging from .39 to .60 for the total scale.

**Table 4. Correlation coefficients obtained in item analysis (items translated into English).**

| Factor | Item | Corrected item-total correlation | |
|---|---|---|---|
| | | Correlation with total scale | Correlation with specific subscale |
| **Scene realism** | 1. Reflection in virtual space seemed to be natural. | .58 | .60 |
| | 2. Light and shades in virtual space were realistic. | .60 | .67 |
| | 3. The virtual space seemed to be three-dimensional. | .43 | .51 |
| | 4. Coloring in virtual space appeared to be natural. | .57 | .61 |
| | 5. Proportions of the virtual space were realistic. | .52 | .55 |
| **Audience behavior** | 6. Posture of virtual humans was natural. | .54 | .49 |
| | 7. Gestures of virtual humans was natural. | .60 | .74 |
| | 8. Behavior of virtual humans in virtual space was authentic. | .57 | .68 |
| | 9. Facial expressions of virtual humans were realistic. | .57 | .66 |
| **Audience appearance** | 10. Outfit of virtual humans was adequate. | .39 | .51 |
| | 11. Virtual humans differed concerning their appearance. | .50 | .58 |
| | 12. Virtual humans in their entirety seemed to be authentic for this occasion. | .58 | .51 |
| | 13. Outfit of virtual humans was natural. | .60 | .66 |
| **Sound realism** | 14. Ambience sound intensity in the virtual room was . . . | .07 | - |

Concerning correlations between an item and a subscale, the results also indicate good discrimination: coefficients ranged from .49 to .74. No coefficient was obtained for the sound realism subscale as it has only one item.

We performed a reliability analysis for each study separately. The obtained Cronbach's alpha coefficients are presented in Table 5. The coefficient for the Sound Realism subscale was not calculated because this dimension consists of only one item.

The obtained Cronbach's alpha coefficients can be considered satisfactory. For the combined scale, alpha ranges from .82 (study C) to .90 (study D). As was predicted, the coefficients of the subscales are lower than of the combined scale, ranging from .66 (audience appearance in study C) to .87 (audience behavior in study E).

## Evidence based on internal structure—The factorial structure of the VR Realism Scale

The tested four-factor model (Scene Realism, Audience Appearance, Audience Behavior, Sound Realism) yields a decent fit: TLI = .965 (good, expected above .95), CFI = .973 (good, expected above .95), RMSEA = .084 (mediocre, expected below .08), SRMR = .076 (good, expected below .08). The incremental indexes yield an acceptable fit. However, the results show a discrepancy between the RMSEA and SRMR indexes, and the RMSEA index did not

**Table 5. Cronbach's Alphas of the VR Realism Scale and its dimensions.**

| Dimension | Number of items | Cronbach's alpha | | | | | |
|---|---|---|---|---|---|---|---|
| | | Study A | Study B | Study C | Study D | Study E | Study F |
| **Combined scale** | 14 | .88 | .85 | .82 | .90 | .88 | .84 |
| **Scene Realism** | 5 | .80 | .75 | .67 | .81 | .79 | .75 |
| **Audience Behavior** | 4 | .82 | .85 | .79 | .86 | .87 | .81 |
| **Audience Appearance** | 4 | .73 | .74 | .66 | .86 | .78 | .68 |
| **Sound Realism** | 1 | – | – | – | – | – | – |

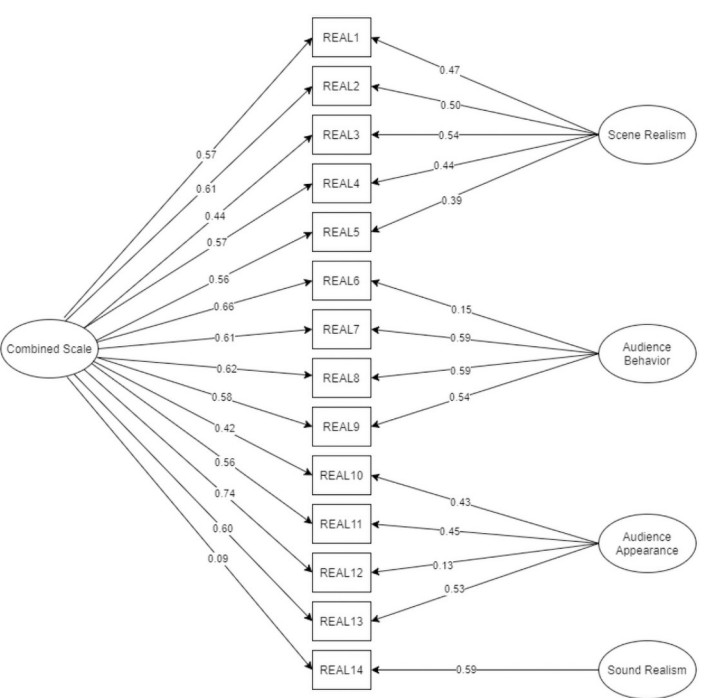

**Fig 2. The four-factor model with standardized factor loadings.**

meet the set criteria. We refer to these results in the discussion section. The obtained factor loadings are presented in Fig 2.

Considering the unsatisfactory fit of the four-factor model, we decided to test another model for comparison. As the latent variables in the four-factor model remained highly correlated with each other (except for the Sound Realism subscale, see Fig 1), we came to the conclusion that testing a bifactor model is justified. The new model consists of four factors (Scene Realism, Audience Appearance, Audience Behavior, Sound Realism) and a general factor that is impacted by all the items of the scale. Orthogonal rotation was used to rule out correlation between factors.

The bifactor model (see Fig 2) yields a satisfactory fit: TLI = .971 (good, expected above .95), CFI = .98 (good, expected above .95), RMSEA = .076 (fair, expected below 0.08), SRMR = .067 (good, expected below 0.08). All the fit indexes meet the assumed criteria; therefore, it can be stated that the observed data confirm the theoretical structure of the VR Realism Scale. The standardized factor loadings are presented in Fig 3. Since the bifactor model yields a satisfactory fit, we present further results for simulation realism treated as a result of the combined scale and for each subscale separately.

**Measurement invariance.** A comparison of the German (n = 181) and Polish (n = 274) groups showed partial configural invariance (Table 6) because there is a discrepancy between the CFI and RMSEA indexes. The RMSEA index yields a satisfactory fit (RMSEA = .073, expected below 0.08) but CFI does not meet the assumed criteria (CFI = .90, expected above .95). We decided to perform subsequent tests. The model for metric invariance yields a very similar fit. The level of CFI (CFI = .89) slightly decreases but RMSEA yields a satisfactory fit (RMSEA = .073). In the next step, we tested scalar invariance. The results (Table 6) could not support evidence for scalar invariance. As is presented in Table 6, we can observe a substantial

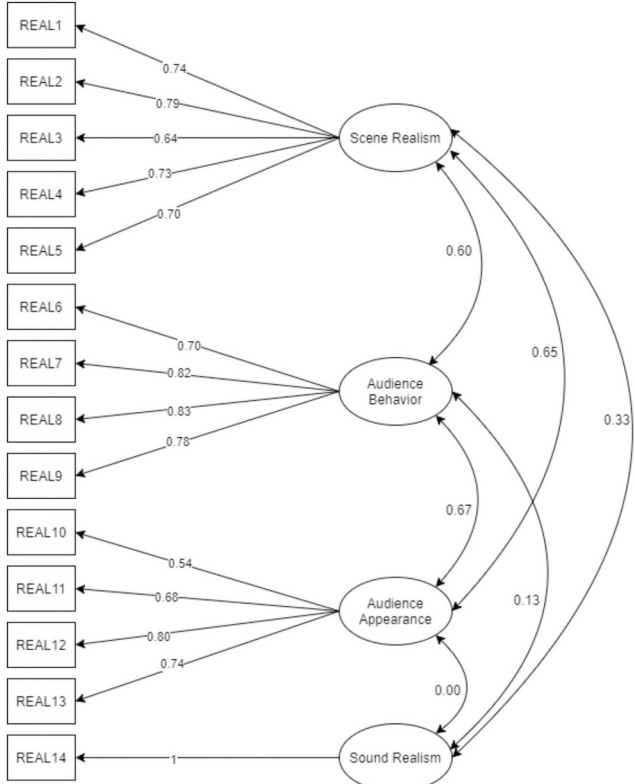

**Fig 3. The bifactor model with standardized factor loadings.**

deterioration of the fit indexes (CFI, RMSEA). It seems that these results indicate non-invariance.

In the next step, we explored which items are non-invariant. Therefore, we analyzed the modification index, which shows that there will be an improved fit if we allow item 12's loading to vary freely. We determined that setting item 12's loading to vary freely across groups could significantly decrease chi$^2$. We decided to verify the non-invariance of item 12. After modification, the fit indexes were slightly improved (Table 6, "partial scalar invariance" row); however, these results cannot support evidence for scalar invariance. As scalar invariance was not verified, we decided to not test residual invariance.

**Table 6. Summarizing the measurement invariance analysis.**

| Model | chi$^2$ | Df | CFI | RMSEA | BIC | AIC | ΔChi$^2$ | Δdf | ΔCFI | ΔRMSEA |
|---|---|---|---|---|---|---|---|---|---|---|
| Configural invariance | 319.19* | 144 | .90 | .073 | 16,159 | 15,772 | - | - | - | - |
| Metric invariance | 340.85* | 154 | .89 | .073 | 16,120 | 15,772 | 21.66 | 10 | .01 | .000 |
| Scalar invariance | 639.66** | 164 | .72 | .113 | 16,358 | 16,053 | 298.81 | 10 | .17 | .040 |
| Partial scalar invariance | 528.91** | 162 | .78 | .100 | 16,258 | 15,945 | 188.06 | 8 | .12 | 0.027 |

*$p < .05$.

**$p < .001$.

## Evidence based on relations to other variables—Correlation analysis

The results of the performed correlation analysis are presented in Table 7. The obtained correlation coefficients are lower than was expected. Simulation realism (understood as the general factor score) correlates weakly and positively with immersion ($r = .35$, $p < .001$). Also observed were a weak positive correlation between simulation realism and spatial presence ($r = .41$, $p < .001$) and a moderate positive correlation with the realness aspect of presence ($r = .57$, $p < .001$). No significant correlation between simulation realism and the involvement aspect of presence was found ($r = .08$, $p = .418$). Simulation realism is also related to all aspects of copresence ($r$ ranging from .31 to .42, $p < .001$). Surprisingly, the highest correlation coefficient was obtained for scene realism and classical aesthetics ($r = .71$, $p < .001$). As was predicted, simulation realism correlates neither with flow state nor with needs satisfaction, except for autonomy need satisfaction ($r = .34$, p $< .001$). We will refer to these results in the Discussion section.

## Evidence based on relations to other variables—Realism and emotions

The high correlation obtained between simulation realism and classical aesthetics may indicate the sensori-emotional value of perceived realism. We decided to determine whether aesthetic experience is an underlying mechanism by which simulation realism influences positive emotions. In order to do this, we performed a mediation analysis (for a visualization of the tested model, see Fig 4).

We performed a series of simple mediation analyses. In our models, the independent variable was simulation realism (understood as the result of the combined scale and each subscale). Classical aesthetics was a mediator variable. As dependent variables, we used two subscales (joy and love) from the scale of six basic emotions. Also, we tested the model with valence as a dependent variable.

For the combined scale (as an independent variable) we found a significant indirect effect between simulation realism and love, mediated by classical aesthetics (beta = .17 95% = [.03, .30]). However, in this case, none of the direct paths (a, b, c') was statistically significant, which may indicate the suppressed role of classical aesthetics. Analysis did not reveal any significant effects for joy or valence (see Table 8 for the coefficients and confidence intervals).

In the next step, we performed a mediation analysis for each subscale. In the case of scene realism, the analyses performed did not reveal any statistically significant effects for joy, love, and valence.

For audience behavior (as an independent variable), the analysis revealed significant indirect effects for joy (beta = .09, 95% CI = [.01, .19]), love (beta = .13, 95% CI = [.04, .24]) and valence (beta = .10, 95% CI = [.01, .21]). Likewise, the assumptions of mediation analysis were not fulfilled for the combined scale: only the indirect effects were statistically significant (see Table 8 for coefficients and confidence intervals). Similar significant indirect effects were found for audience appearance. In this case, the analysis revealed suppression effects for joy (beta = .07, 95% CI = [.01, .16]), love (beta = .12, 95% CI = [.04, .21]) and valence (beta = .07, 95% CI = [.01, .18]).

## Evidence based on test content—Sensitivity of the scale

In order to test the sensitivity of the scale, the results obtained in study D and study E were compared. As was mentioned in the Materials and Methods section, these two studies were parts of a larger research program with the same participants, procedures, and tasks. The virtual environments used in these studies differed only in terms of the graphics and sound. Sounds were made more adequate in terms of volume (louder) and content. Sounds of road traffic and people wailing and moaning were added. Some models were also improved: more

**Table 7. Correlations with similar measures.**

| Variable | Immersion | Presence | | | Co-presence | | | | Aesthetics | | Player Needs Satisfaction | | | Flow |
|---|---|---|---|---|---|---|---|---|---|---|---|---|---|---|
| | Immersion | Spatial Presence | Involvement | Realness | Presenters reaction to virtual agents | Perceived virtual agents' reaction | Impression of interaction possibilities | Other people | Classical Aesthetics | Expressive Aesthetics | Competence Need | Autonomy Need | Relatedness Need | Flow |
| Combined Scale | .32** | .41** | .08 | .57** | .31** | .35** | .42** | .38** | .69* | .51* | .01 | .37** | .04 | .11 |
| Scene realism | .27** | .43** | .13 | .53** | .31** | .35**. | .38** | .35** | .71** | .47** | .03 | .37** | .05 | .10 |
| Audience behavior | .37** | .37** | .14 | .54** | .29** | .38** | .49** | .42** | .55** | .50** | .01 | .32** | .06 | .10 |
| Audience Appearance | .17* | .31** | -.03 | .40** | .19 | .19 | .26** | .26** | .59** | .35** | -.01 | .25** | -.04 | .10 |
| Sound realism | .06 | .05 | -.12 | .29** | .21* | .14 | .14 | .11 | .18 | .21* | .04 | .13 | .04 | .01 |

*$p < .05$.
**$p < .01$.

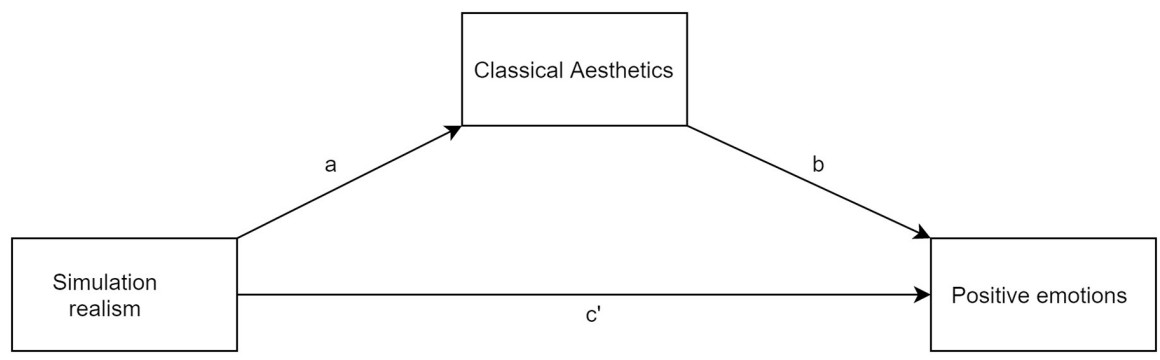

**Fig 4. Simple mediation model.**

details were added to the inside of the car and its doors were made thicker. An animation of an opening and closing mouth while checking airways was added. The virtual agents' behavior was improved: for example, one of the virtual victims would lose consciousness (fall to the ground) 40s after engaging in an interaction with the virtual agent. An example of these improvements is shown in Fig 5. We expected an increase in scene realism between these two studies because more textures had been added to the virtual objects. Also, we predicted that audience appearance and behavior would be assessed as better because of the new animation of the opening and closing mouth. These two aspects of realism might also have been influenced by the adding of a situation in which one of the victims loses consciousness. We expected that adding moaning and yelling sounds would improve sound realism.

To verify whether the VR Realism Scale is sensitive to small changes in a simulation, a *t*-test for dependent samples was conducted. The results of the analysis are presented in Table 9. Significant differences were also detected for the combined scale and all four aspects of simulation realism. The effect sizes for combined scale, scene realism, audience behavior realism,

**Table 8. Standardized regression coefficients for the relationship between simulation realism and positive emotions mediated by classical aesthetics.**

| Independent variable | Dependent variable | Standardized regression coefficients | | | | | | | |
|---|---|---|---|---|---|---|---|---|---|
| | | path a | | path b | | direct effect–path c' | | indirect effect–path ab | |
| | | beta | 95% CI | beta | 95% CI | beta | 95% CI | beta | 95% CI |
| **Combined scale** | **Joy** | .59 | .71, 1.22 | .16 | -.07, .46 | .15 | -1.16, .72 | .10 | -.03, .22 |
| | **Love** | .59 | .71, 1.22 | .29 | .07, .48 | .06 | -.24, .43 | .17 | .03, .30* |
| | **Valence** | .59 | .71, 1.22 | .18 | -.07, .63 | .11 | -.30, .84 | .11 | -.01, .25 |
| **Scene Realism** | **Joy** | .66 | .79, 1.22 | .22 | -.04, .54 | .05 | -.34, .54 | .14 | -.01, .29 |
| | **Love** | .66 | .79, 1.22 | .24 | -.00, .44 | .14 | -.14, .53 | .16 | -.01, .32 |
| | **Valence** | .66 | .79, 1.22 | .18 | -.10, .65 | .10 | -.35, .79 | .12 | -.04, .28 |
| **Audience behavior** | **Joy** | .43 | .29, .68 | .20 | -.01, .47 | .13 | -.10, .44 | .09 | .01, .19* |
| | **Love** | .43 | .29, .68 | .31 | .10, .47 | .05 | -.15, .26 | .13 | .04, .24* |
| | **Valence** | .43 | .29, .68 | .22 | .03, .65 | .05 | -.25, .44 | .10 | .01, .21* |
| **Audience appearance** | **Joy** | .35 | .25, .78 | .21 | .02, .48 | .12 | -.14, .54 | .07 | .01, .16* |
| | **Love** | .35 | .25, .78 | .35 | .14, .50 | -.04 | .-.31, .21 | .12 | .04, .21* |
| | **Valence** | .35 | .25, .78 | .21 | .02, .63 | .09 | -.23, .65 | .07 | .01, .18* |

* *significant* effect.

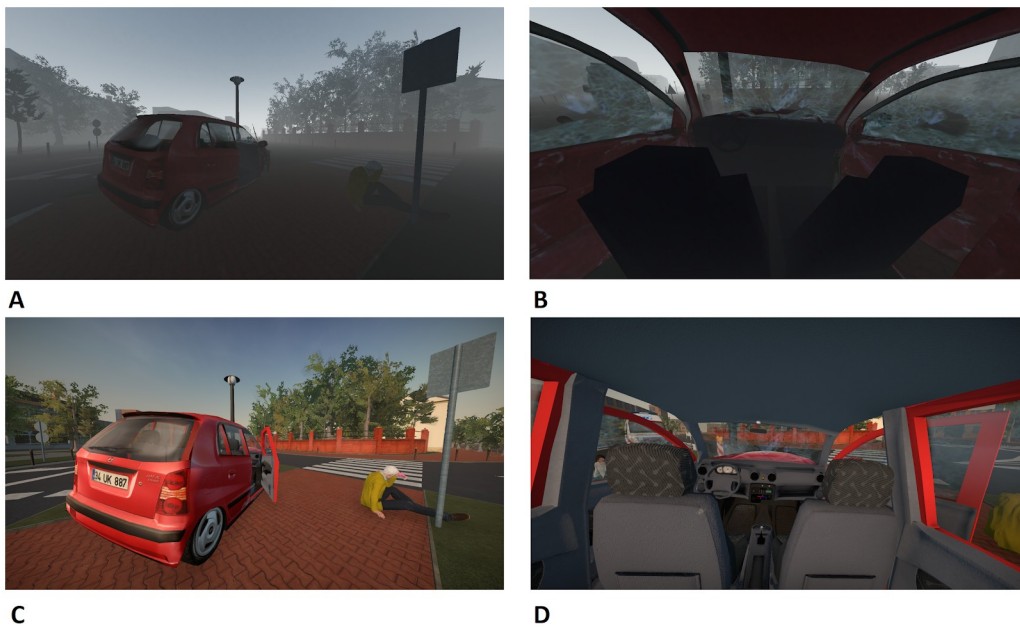

**Fig 5. An example of some of the improvements made to the simulator.** Panels A and B are from the earlier version; panels C and D are from the second, improved version.

audience appearance realism and sound realism (*d* ranges between .20 and .42) should be considered small [91].

## Discussion

The presented analyses aimed to validate the psychometric properties of the scale, confirm its sensitivity, and explore the relationships with other IVE characteristics. To achieve these aims, we used data from different sources.

We considered the confirmation of the original scale structure to be essential as, to the best of our knowledge, we are the first research team to use the Polish version of the scale. The factorial structure of the Polish version of the VR Realism Scale was compared to the original four-factor structure. This model has a satisfactory fit except for one absolute index: RMSEA. Although RMSEA is one of the most popular fit indexes, it may not yield accurately when fitting ordinal factor analysis [92]. We decided to apply diagonally weighted least squares

**Table 9. *T*-test results.**

|  | Study D | | Study E | | t-test | | | Effect size |
|---|---|---|---|---|---|---|---|---|
|  | *M* | *SD* | *M* | *SD* | *t* | *df* | *p* | *Cohen d* |
| **Combined Scale** | 0.17 | 0.67 | 0.39 | 0.61 | -3.48 | 108 | .001** | .34 |
| **Scene Realism** | 0.20 | 0.74 | 0.36 | 0.69 | -2.11 | 108 | .037* | .21 |
| **Audience Behavior** | -0.10 | 0.88 | 0.18 | 0.94 | -3.24 | 108 | .002** | .30 |
| **Audience Appearance** | 0.58 | 0.88 | 0.78 | 0.71 | -2.13 | 108 | .036* | .20 |
| **Sound Realism** | -0.57 | 0.92 | -0.13 | 0.88 | -4.42 | 108 | .001** | .42 |

* *p* < .05.
** *p* < .01.

estimation because our endogenous variable was derived from answers on a Likert scale and was therefore categorical. In this case, SRMR may be more accurate when assessing the degree of misfit [92].

In this model, three of the four latent variables remain correlated (from $r = .59$ to $r = .69$, see Fig 1) with each other. For this reason, we also decided to test the bifactor model. In this way, the similarity to the original version of the scale is preserved but the general factor is loaded directly by items. The bifactor model allows researchers to calculate results on either subscales or one combined scale [93]. The fit of this model is satisfactory. Compared to the four-factor model, all indexes are slightly improved (RMSEA$_{decrease}$ = .005, SRMR$_{decrease}$ = .007, CFI$_{increase}$ = .006, TLI$_{increase}$ = .004). In our opinion, the obtained CFA coefficients provide sufficient evidence to support the validity of the internal structure of the scale.

In our opinion, the bifactor model is exceptionally useful in the case of simulation realism. As we mentioned before, rather than adding more textures to objects or agents, the consistency between various aspects of realism is crucial for the virtual experience [94]. Therefore, evaluating simulation realism aspects separately may be not the correct approach. However, calculating the combined scale score may lead to loss of variance, therefore the discrepancy between simulation realism aspects may be unnoticed. Additionally, there are simulations without virtual agents, and in such cases the combined scale score may be considered meaningless. Bifactor models are used for the assessment of a construct that is treated as one-dimensional and multidimensional at the same time [95]. Therefore, we recommend comparing the means obtained in the subscales in the first step. Then, if the means are comparable, we recommend calculating the combined scale score.

As our analyses consider a validation of an existing tool, we tested measurement invariance to determine whether the German and Polish versions are comparable across groups; however, the results seem to be inconclusive. RMSEA yields satisfactory fits for configural and metric invariance, but the CFI indexes are slightly too low to be considered a good fit. Additionally, interpretation of comparative indexes, namely AIC and BIC, indicates that metric invariance may be supported (a decrease in comparison to configural invariance). We could not obtain any evidence for scalar invariance, even though we set item 12 to vary freely across groups; therefore, the results may indicate scalar non-invariance. Summarizing, the structure of the scale is the same in both versions. Moreover, it seems that the items in the Polish and German versions of the scale are understood in the same way in both groups. Scalar non-invariance may be the result of one group's tendency to systematically over- or under-respond to the questionnaire [96]. Scalar non-invariance may also be due to construct bias or method bias [97]. The development and validation processes of the German and Polish versions of the VR realism scale differ in terms of participants' language (German vs Polish) and the technology used (CAVE vs HMD). The translation process was conducted in accordance with the state of the art (i.e., including back-translation by professional translators and a bilingual person; linguistic consultation; approval of the original scale author; pilot study). Additionally, our results may support evidence for partial metric invariance. Therefore, it seems to us that scalar non-invariance may be a result of the different technology used during the validation process, although we do not have data that could verify this assumption. In order conclude that the Simulation Realism Scale provides different results depending whether a CAVE or an HMD is being assessed, an experiment with two groups (CAVE vs HMD) that speak the same language should be conducted.

We obtained evidence based on relations to other variables by performing correlation analysis that included variables which are related to simulation realism (presence, immersion, co-presence, aesthetics) on the basis of theoretical assumptions, and variables which are important for the virtual experience but are not necessarily related to simulation realism (flow,

players' needs satisfaction). The correlations with both immersion and presence are surprisingly low. We expected that simulation realism, as an aspect of fidelity, would be strongly related to immersion. Perhaps immersion is more strongly related to other aspects of fidelity, namely display and interaction fidelities. If so, a weak connection between simulation realism and immersion may support the thesis that simulation realism is a purely perceptual experience, whereas immersion (i.e. fidelity) is a result of technological capabilities [9]. In turn, the sense of presence may be a much more complex cognitive process [59] that cannot be explained by just one factor, namely simulation realism. Nevertheless, we expected a much higher correlation between simulation realism and one aspect of presence, namely realness, as these two variables seem to overlap. Perhaps this moderate correlation (r = .53) between scene realism and realness supports the existence of the method effect [98]. Our results are in line with evidence that indicates inconsistency in the relation between presence and realism [33, 34]. It seems that sense of presence is not only dependent on perceiving the virtual environment but also on the ability to take action [99]. In the light of our results, it may be concluded that simulation realism is not enough to evoke a strong sense of 'being there'.

In the case of the relation between simulation realism and aspects of co-presence, the Pearson's *r* coefficient obtained for the 'audience behavior' subscale is relatively higher than that for 'scene realism' and 'audience appearance'. These predictions are in line with research [35–37] and theoretical predictions [25] that emphasize the importance of behavioral realism over appearance realism. The fact that co-presence correlates more highly with audience behavior realism than with other aspects of simulation realism may support the congruent validity of the scale as well as the quality difference between subscales.

As we predicted, the relation between realism and flow and the relation between realism and players' needs satisfaction can be considered as discriminant evidence, except for the autonomy need. We did not expect a correlation between simulation realism and autonomy need satisfaction. This result could indicate that realism may be somehow involved in intrinsic motives. The satisfaction of autonomy need is enhanced by providing a plurality of choices and a sense of freedom [54]. Perhaps a complex virtual environment with many interaction possibilities can satisfy this need and thus increase users' well-being. In this sense, realism is not only limited to the visual aspect: the fidelity of the physical world in terms of the available actions is also important.

Based on the results of correlation analysis and the connection between simulation realism and aesthetics, we decided to determine whether the scale scores have one more consequence: the ability of realism to evoke pleasure, namely positive emotions in this case. We discovered a hidden relation between realism and positive affect (the emotions of joy and love), and between realism and valence. In the tested models, there was no significant relation between simulation realism and any of the tested positive emotions. Nevertheless, we tested whether controlling for aesthetic assessment changes this relation. We found that classical aesthetics is in fact a suppressor of the relation between realism and positive affect (the emotions of joy and love). Interestingly, classical aesthetics suppressed the influence of simulation realism on positive affect but only in the case of social aspects of realism (subscales: audience behavior, audience appearance). It is worth noting that the mediation analysis did not meet the traditional assumptions established by Baron and Kenny [87]. However, more recent works emphasize that calculating indirect effects is allowed even if there is no direct influence of the independent variable on the dependent variable [88]. Bearing in mind the fact that we did not follow the traditional approach in this part of our analyses, we recommend further exploration regarding the relationship between positive emotions and realism. We believe that this relationship is of particular importance as many simulations aim to evoke negative affect for the purposes of training [5] or therapy [8]. Therefore, if high simulation realism induces positive emotions,

this issue should be carefully considered when designing virtual environments whose goal is to evoke emotions of negative valence, such as tools for anxiety treatment.

To obtain evidence based on test content and to test the sensitivity of the scale, we verified how improving the design of the virtual models influences the assessment of simulation realism. The participants were asked twice at an interval of two months to complete the same VR task. The second time, the simulation was slightly improved in terms of the quality of textures. The analyses show an increase in the assessment of simulation realism in terms of scene realism, audience behavior, audience appearance, sound realism, and general impression. The obtained results may indicate that users are sensitive to small changes in simulation graphics and that the simulation realism measured by the VR Realism Scale is operationalized adequately. Furthermore, these results show the ability of the scale to capture even small changes in the perception of a simulation, which makes it a promising tool for both developers and scientists.

Regarding evidence based on response processes, we did not follow recommendations such as using eye-tracking, interviews or focus groups [100]. However, during the translation process we conducted a pilot study (see *Translation process* subsection) to verify that the items of the scale are understandable and sound natural. In the final version of Polish scale, we included all participants' comments. During our research program, none of the participants (n = 720) reported that the items were unclear or difficult to answer. Moreover, all items are affirmative single sentences, which makes them easily to process [101]. We believe that the arguments given are sufficient to be considered as evidence that is based on response processes.

## Limitations and future directions

We agree with the authors of the original questionnaire that the issue of sound realism should be addressed in more detail [11] in the future. In fact, we decided to include it in the analyses because our goal was to test the model as it was created. However, for the findings summarized below, we believe that it is necessary to consider the possibility of removing the audio realism item from the analyses when designing any future study. This single item is formulated counter-intuitively (the "best" answer is located in the middle of the scale, unlike all the other items), which may have distorted the analysis. In the case of the four-factor model, sound realism does not correlate with any other latent variable. In turn, in the bifactor model we observed a lack of influence of the audio realism item on this latent variable. The content of this item relates only to audio volume, which cannot be considered an accurate measurement of the more complex phenomenon of the sound aspect of realism. Considering the above, our recommendation for future research is to drop the single sound item from analysis. On the other hand, we do not believe that omission of this item should be mandatory because, at least until the publication of a validated tool for measuring sound realism, gathering information with this item may provide a substitute for information about this aspect of realism. Including this item in analysis also allows for standardization of methods and increases the comparability of studies. Future work should also address the development of a more accurate scale to measure sound realism. The sheer volume level is something that can be easily corrected by the experimenter during the experiment and by the user during daily use, therefore it should not lead to disruptions in perceived realism. Perhaps, instead of measuring the level of adequateness of volume, developing items concerning the emotional prosody of virtual agents would be more beneficial.

In its present form, the scale consists of items that concern only the audio-visual and audio aspects of simulation realism. However, these are not the only ones that may be experienced in an IVE. Modern simulators can provide users with haptic feedback, smell or even radiation

[102]; thus, in the future, measurement of simulation realism should also include other sensory modalities.

We were not able to obtain evidence based on the consequences of testing. More research needs to be done to determine what can be predicted on the basis of a scale score. We believe that two directions are worthy of further research. Firstly, we discovered a hidden relation between simulation realism and positive affect. Future research should include identification of the underlying mechanism of this relationship. Secondly, the influence of simulation realism on users' behavior during VR sessions should be tested. The question of whether users' assessment of realism changes the way they act needs to be taken into consideration.

## Conclusions

In the paper we present a validation of the VR Simulation Realism questionnaire. Our work is beneficial for both academia and practical applications. When it comes to science per se, our research is a step towards better understanding of the virtual experience. We provide analyses which directly indicate relationships between realism and the psychological characteristics of an effective simulation. Furthermore, we show the definitional boundaries of realism, and we confirm the structure, usefulness and sensitivity of this scale. We believe that this paper provides evidence that the VR Realism Scale is a well-tested tool that can be used to measure one of the crucial aspects of IVEs, namely realism; therefore, it should be useful in both science and VR development. However, our work has some limitations: the measurement of only visual and auditory modalities and the lack of evidence based on the consequences of testing. These limitations should be considered as future directions of research.

## Supporting information

**S1 Appendix. Polish version of the VR Realism Scale.**
(PDF)

**S1 File. Procedures and measures.**
(PDF)

**S1 Dataset. Data used for analysis.**
(XLSX)

## Acknowledgments

The authors would like to thank Krzysztof Rębilas for helping with the data acquisition, dr Gabriela Czarnek for sharing her methodological knowledge in the process of designing the procedure, Konrad Klocek and the whole Nano Games programming team for creating the VR simulator used in the studies.

## Author Contributions

**Conceptualization:** Natalia Lipp, Radosław Sterna, Natalia Dużmańska-Misiarczyk, Paweł Strojny.

**Data curation:** Natalia Lipp, Natalia Dużmańska-Misiarczyk, Agnieszka Strojny.

**Formal analysis:** Natalia Lipp, Natalia Dużmańska-Misiarczyk.

**Funding acquisition:** Agnieszka Strojny, Paweł Strojny.

**Investigation:** Natalia Lipp, Radosław Sterna, Natalia Dużmańska-Misiarczyk.

**Methodology:** Agnieszka Strojny, Paweł Strojny.

**Project administration:** Agnieszka Strojny, Paweł Strojny.

**Supervision:** Sandra Poeschl-Guenther, Paweł Strojny.

**Visualization:** Natalia Lipp.

**Writing – original draft:** Natalia Lipp, Radosław Sterna, Natalia Dużmańska-Misiarczyk.

**Writing – review & editing:** Natalia Lipp, Radosław Sterna, Natalia Dużmańska-Misiarczyk, Agnieszka Strojny, Sandra Poeschl-Guenther, Paweł Strojny.

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
