## [Decision Letter · Decision Letter 0]

14 May 2021

PONE-D-21-05975

VR Realism Scale – revalidation of contemporary VR headsets on a Polish sample

PLOS ONE

Dear Dr. Lipp,

Thank you for submitting your manuscript to PLOS ONE. After careful consideration, we feel that it has merit but does not fully meet PLOS ONE’s publication criteria as it currently stands. Therefore, we invite you to submit a revised version of the manuscript that addresses the points raised during the review process.

We look forward to receiving your revised manuscript.

Kind regards,

Meng-Cheng Wang

Academic Editor

PLOS ONE

Journal Requirements:

3. Throughout your manuscript, we noticed several instances of statistical values that are showing up incorrectly and/or are replaced with a blank box, e.g., ln 312. Please correct these in your resubmission.

4. Thank you for providing the following Funding Statement: 

[The work of RS was financed from the science budget (https://www.gov.pl/web/edukacja-i-nauka/diamentowy-grant) for the years 2019–2023 as a research project, ‘The impact of virtually generated characters: human aspects of bots’ (project number: DI2018 015848), under the Diamond Grant program granted by the Ministry of Science and Higher Education of Poland.

Studies A–E were supported by the Polish National Centre for Research and Development (https://www.gov.pl/web/ncbr) under the Widespread Disaster Simulator grant – research and preparation for implementation (project number: POIR.01.01.01-00.0042/16; the Smart Growth Operational Programme, sub-measure 1.1.1. Industrial research and development work implemented by enterprises) received by Nano Games sp. z o.o. AS, PS, NDM, NL, RS received salaries from Nano Games sp. z o.o.

Study F was supported by the Polish National Science Centre (https://www.ncn.gov.pl/) under the grant ‘Wirtualna śmierć realne konsekwencje - efekty śmierci bohatera w grze wideo w świetle Teorii opanowania trwogi’ [ang. Virtual death – real consequences. The effect of virtual agent’s death according to the terror management theory] received by AS (project number: DEC-2018/02/X/HS6/01426 ).

 The funders had no role in study design, data collection and analysis, decision to publish, or preparation of the manuscript.]. 

We note that one or more of the authors is affiliated with the funding organization, indicating the funder may have had some role in the design, data collection, analysis or preparation of your manuscript for publication; in other words, the funder played an indirect role through the participation of the co-authors.

If the funding organization did not play a role in the study design, data collection and analysis, decision to publish, or preparation of the manuscript and only provided financial support in the form of authors' salaries and/or research materials, please review your statements relating to the author contributions, and ensure you have specifically and accurately indicated the role(s) that these authors had in your study in the Author Contributions section of the online submission form. Please make any necessary amendments directly within this section of the online submission form.  Please also update your Funding Statement to include the following statement: “The funder provided support in the form of salaries for authors [insert relevant initials], but did not have any additional role in the study design, data collection and analysis, decision to publish, or preparation of the manuscript. The specific roles of these authors are articulated in the ‘author contributions’ section.”

If the funding organization did have an additional role, please state and explain that role within your Funding Statement.

Please also provide an updated Competing Interests Statement declaring this commercial affiliation along with any other relevant declarations relating to employment, consultancy, patents, products in development, or marketed products, etc.  

6. We note that your paper includes detailed descriptions of car licence plate number (Figure 4). As per the PLOS ONE policy (http://journals.plos.org/plosone/s/submission-guidelines#loc-human-subjects-research) on papers that include identifying, or potentially identifying, information, the individual(s) or parent(s)/guardian(s) must be informed of the terms of the PLOS open-access (CC-BY) license and provide specific permission for publication of these details under the terms of this license. Please download the Consent Form for Publication in a PLOS Journal (http://journals.plos.org/plosone/s/file?id=8ce6/plos-consent-form-english.pdf). The signed consent form should not be submitted with the manuscript, but should be securely filed in the individual's case notes. Please amend the methods section and ethics statement of the manuscript to explicitly state that the patient/participant has provided consent for publication: “The individual in this manuscript has given written informed consent (as outlined in PLOS consent form) to publish these case details”.

Reviewers' comments:

Reviewer's Responses to Questions

**Comments to the Author**

1. Is the manuscript technically sound, and do the data support the conclusions?

Reviewer #1: Yes

Reviewer #2: Partly

2. Has the statistical analysis been performed appropriately and rigorously? 

Reviewer #1: Yes

Reviewer #2: Yes

3. Have the authors made all data underlying the findings in their manuscript fully available?

Reviewer #1: Yes

Reviewer #2: Yes

4. Is the manuscript presented in an intelligible fashion and written in standard English?

Reviewer #1: Yes

Reviewer #2: Yes

5. Review Comments to the Author

Reviewer #1: Thank you for the opportunity to review “VR Realism Scale — revalidation of contemporary VR headsets on a Polish sample” in the PLOS ONE. This study present validation of the VR Simulation Realism Scale in a sample of Polish participants. I appreciate the authors’ effort of examining psychometric properties of the VR Simulation Realism Scale through five experiments and one online survey, and I have several comments below that the authors may consider.

1. The authors do not make a strong case for the importance of the study (examine psychometric properties of the VR Simulation Realism Scale in a sample of Polish speaking participants). The Introduction does not discuss why this is important.

2. Although the authors discuss immersion, presence and co-presence, the concepts of flow and realism need also properly describe in the Introduction section.

3. In my opinion, I recommend that the authors add the more description of the relationship between VR and criterial measures (e.g., other IVE characteristics) in Introduction section. In particular, the prediction model (i.e., mediation analysis) in the current study.

4. The data analysis strategy is missing in the Material and Methods section, and the related criteria (e.g., internal consistency and CFA) should be described in the Material and Methods (data analysis) not in the Results section.

5. Before conducting the reliability and validity analysis, I suggest that the authors add the item analysis of VR Simulation Realism Scale.

6. Both the model fit of CFA and Bifactor were acceptable, while the factor loadings of several items (e.g., items 1, 6, 10, 14) were unsatisfactory. In general, I’m a little confused why the Sound Realism factor has only one item. Typical, a latent factor requires at least three items to be effectively measured.

7. According to the CFA and Bifactor, the factor loadings of items 1, 6, 10 and 14 were unsatisfactory. The authors need add some description in the Discussion section.

8. It is suggested that the authors further explain and elaborate the relationship between VR and relevant variables (e.g., correlation analysis, mediation analysis) in the Discussion section.

9. The format specification of references.

Other Minor:

1.The expression of IVE and IVEs in the manuscript needs to be consistent.

2. In the table 4, the number of items for Scene Realism factor should be 5.

Reviewer #2: Thanks for the opportunity to review this paper. This study contributes to the virtual experience research by multiple studies with mixed methods. A lot of work has been done to validate the VR Realism Scale and reveal its psychometric properties in Polish context. It is valuable and meaningful. However, I have several concerns, which need to be addressed to improve this paper.

1. It is not easy to follow what your research question is and why it is important in “Introduction”. It seemed that you mentioned it in “Aims and rationale behind our study”, which confused me a little bit regarding the relationship between these two parts. Was the part of “Aims and rationale behind our study” included in the part of “Introduction”? It would be better to reorganize Introduction, with the aim to clarify your research question and why it needs to be addressed logically.

2. As validation of the VR Realism Scale in a different context (i.e., Polish) from where it was developed, is one of the main goals of this paper, measurement invariance across cultures is the key point. What measures did you take to ensure the same meaning of the obtained Polish version scale as that of the German version scale? Who were responsible for back-translating, or comparing the two versions? What were their qualifications for doing so? How did you decide the final version? Could you please provide more details about the scale translation process?

3. A related question, what changes did you make to the original scale? You introduced that “examine the psychometric properties of the scale in the context of state-of-the-art VR technology and modern IVEs, as CAVES and modern headsets differ greatly in terms of the characteristics of the stimuli they deliver (e.g., Mestre, 2017)”. As the technological environments focused are different, what changes did you do for the original one to fit the situation of VR headsets you were specific on?

4. I am not clear about the implication of the bi-factor model result. Is it possible to support the existence of the method effect, as all items were reported by the participates self? And if you could justify that bi-factor model support the one combined structure of simulation Realism, how should we measure this construct, as one general dimension or as multi-dimensions? A related question, why did you treat it as both unidimensional and multi-dimensional in the following analyses?

5. I agree with that it is necessary to examine the relations of simulation realism to other variables. But why did you theorize a mediation model wherein aesthetics as the mediator between simulation realism and positive emotions? Could you please provide more powerful arguments on this study, and explain why this mediation model is needed or helpful to achieve your goal of investigating its relations?

Hope these comments are useful for you.

6. PLOS authors have the option to publish the peer review history of their article (what does this mean?). If published, this will include your full peer review and any attached files.

Reviewer #1: No

Reviewer #2: No

---

## [Author Response · Author response to Decision Letter 0]

20 Jul 2021

Authors’ responses to the Reviewers’ comments

First of all, we would like to thank the Reviewers for all their valuable comments. Thanks to them we could improve our text. Detailed answers to the comments are provided below.

Journal Requirements 

Thank you for spotting the formatting errors. We worked on solving that issue. We changed the format of the references and we updated first page of the manuscript.

The information about participant consent was inserted into the main body of the text (in the Procedures section). 

3. Throughout your manuscript, we noticed several instances of statistical values that are showing up incorrectly and/or are replaced with a blank box, e.g., ln 312. Please correct these in your resubmission. 

Thank you for pointing that out. The symbols were corrected in the text. As we were not sure which symbols were visible on your side, we decided to replace the Greek letters with word descriptions to make sure that they will be visible. We hope that this solution is acceptable. 

4. We note that one or more of the authors is affiliated with the funding organization, indicating the funder may have had some role in the design, data collection, analysis or preparation of your manuscript for publication; in other words, the funder played an indirect role through the participation of the co-authors. 

If the funding organization did not play a role in the study design, data collection and analysis, decision to publish, or preparation of the manuscript and only provided financial support in the form of authors' salaries and/or research materials, please review your statements relating to the author contributions, and ensure you have specifically and accurately indicated the role(s) that these authors had in your study in the Author Contributions section of the online submission form. Please make any necessary amendments directly within this section of the online submission form. Please also update your Funding Statement to include the following statement: “The funder provided support in the form of salaries for authors [insert relevant initials], but did not have any additional role in the study design, data collection and analysis, decision to publish, or preparation of the manuscript. The specific roles of these authors are articulated in the ‘author contributions’ section.”

1. The institutions financing the research reported in the manuscript (Ministry of Science and Higher Education of Poland, Polish National Center for Research and Development, Polish National Science Center) are not affiliated with any of the authors; however, they had an indirect impact on research - they decided to finance them on the basis of an assessment of their value by expert researchers paid by these institutions. The authors are affiliated with the commercial institution ((Nano Games sp. z o. o.)) as they are Nano Games employees, but Nano Games is the grant receiver not the funder.

2. Some authors were indeed employed by a company (Nano Games), whose simulator became the test environment for the adapted questionnaire. In fact, the initiative to adapt the tool in question stemmed from the fact that Nano Games needed a validated, Polish-language tool to assess the realism of its products. However, part of the agreement between these parties was the consent to publish the results of scientific research planned under the above-mentioned projects. This consent was made before the research was carried out. Moreover, the authors also undertook to publish the results, as this in turn was declared by Nano Games during the assessment of the financing institution (Polish National Center for Research and Development). Neither Nano Games nor any Nano Games’ employee outside the authors of the manuscript did not play a role in the study design, data collection and analysis, decision to publish (beyond the previously agreed purpose of publication), or preparation of the manuscript. 

We updated our Competing Interests Statement according to PLOS ONE guidelines: “Our commercial affiliation does not alter our adherence to PLOS ONED policies on sharing data and materials” 

We believe that the above explanation and supplementation of the declarations contained in the manuscript truthfully and strictly according to the editor's proposal will resolve all doubts. 

5. We note that your paper includes detailed descriptions of car licence plate number (Figure 4). As per the PLOS ONE policy (http://journals.plos.org/plosone/s/submission-guidelines#loc-human-subjects-research) on papers that include identifying, or potentially identifying, information, the individual(s) or parent(s)/guardian(s) must be informed of the terms of the PLOS open-access (CC-BY) license and provide specific permission for publication of these details under the terms of this license. Please download the Consent Form for Publication in a PLOS Journal (http://journals.plos.org/plosone/s/file?id=8ce6/plos-consent-form-english.pdf). The signed consent form should not be submitted with the manuscript, but should be securely filed in the individual's case notes. Please amend the methods section and ethics statement of the manuscript to explicitly state that the patient/participant has provided consent for publication: “The individual in this manuscript has given written informed consent (as outlined in PLOS consent form) to publish these case details”.

Thank you for your vigilance. The car present on the virtual accident site was purely fictitious and therefore the license plates are fictitious as well and they do not belong to any identifiable person – so no consent could be provided. The numbers were provided in a realistic manner for the purpose of the ecological validity of the virtual environment. 

If you believe that the presence of the numbers could be a problem from the ethical perspective anyway, we propose that we can blur the numbers on the figure, so that they would be unreadable. 

Reviewer 1 

1. The authors do not make a strong case for the importance of the study (examine psychometric properties of the VR Simulation Realism Scale in a sample of Polish speaking participants). The Introduction does not discuss why this is important.

Thank you for this remark. We reorganized the Introduction section and we tired to expand on the rationale behind our study. We give more arguments for the revalidation of the scale in Polish.

2. Although the authors discuss immersion, presence and co-presence, the concepts of flow and realism need also properly describe in the Introduction section.

Thank you for this remark. As it was suggested, the concepts of flow and realism are introduced and defined next to immersion, presence, and co-presence. 

3. In my opinion, I recommend that the authors add the more description of the relationship between VR and criterial measures (e.g., other IVE characteristics) in Introduction section. In particular, the prediction model (i.e., mediation analysis) in the current study. 

As it was suggested, we expanded the description of these relationships, and added our assumptions about relationship between simulation realism, aesthetics and positive affect (the prediction model) in Introduction section. 

4. The data analysis strategy is missing in the Material and Methods section, and the related criteria (e.g., internal consistency and CFA) should be described in the Material and Methods (data analysis) not in the Results section. 

We hope that we addressed your concern. According to the suggestion, all information about conducted analyses and adopted criteria was moved to the ‘Data Analysis’ section.

5. Before conducting the reliability and validity analysis, I suggest that the authors add the item analysis of VR Simulation Realism Scale. 

Thank you for this suggestion. We added the item analysis. We conducted the corrected item total correlation analysis. 

6. Both the model fit of CFA and Bifactor were acceptable, while the factor loadings of several items (e.g., items 1, 6, 10, 14) were unsatisfactory. In general, I’m a little confused why the Sound Realism factor has only one item. Typical, a latent factor requires at least three items to be effectively measured.

Thank you for that remark. It seems to us that there may have been a misunderstanding due to incorrect labeling of the figures. The factor loadings of items 1, 6, 10, 14 are not unsatisfactory, they were not calculated because these items were set to 1 to serve as marker indicators. In confirmatory factor analysis one item (per factor) is always fixed to 1.0 and R package Lavaan automatically chooses the first one. To increase the clarity of our results, we corrected figures and added information about marker indicator to the Results section. 

Concerning single-item subscale of Sound Realism, we were confused about it as well. Nevertheless, we decided to validate the original model as it was originally designed. We discussed this issue in two parts of the manuscript: in introduction (‘Aims of our study’ section) and the discussion.

7. According to the CFA and Bifactor, the factor loadings of items 1, 6, 10 and 14 were unsatisfactory. The authors need add some description in the Discussion section. 

Thank you for this suggestion. We tried to address this issue in the previous answer. 

8. It is suggested that the authors further explain and elaborate the relationship between VR and relevant variables (e.g., correlation analysis, mediation analysis) in the Discussion section. 

Thank you for this remark. We discuss more about these relationships. We tried to find explanations and we added more info about results regarding connections between simulation realism and immersion, simulation realism and presence, and simulation realism and aesthetics.

9. The format specification of references.

Thank you for your vigilance. We corrected the format of the references. 

10. The expression of IVE and IVEs in the manuscript needs to be consistent. 

Thank you for finding the inconsistency. We followed your point and the expressions of the abbreviations were corrected, now the “IVE” abbreviation is used consistently throughout the text. 

11. In the table 4, the number of items for Scene Realism factor should be 5. 

The number was corrected, thank you for your vigilance. 

Reviewer 2 

1. It is not easy to follow what your research question is and why it is important in “Introduction”. It seemed that you mentioned it in “Aims and rationale behind our study”, which confused me a little bit regarding the relationship between these two parts. Was the part of “Aims and rationale behind our study” included in the part of “Introduction”? It would be better to reorganize Introduction, with the aim to clarify your research question and why it needs to be addressed logically.

Thank you for underlining this issue. Indeed, “Aims and rationale behind our study” was meant to be a part of “Introduction” which would make clear the importance and reason of our paper. however, thanks to Reviewer’s comment, we realized that it may not be easy to follow our argumentation. Therefore, we reorganized the “Introduction” section. The new version starts with our main goal and the justification, and then we introduced related variables. We hope that this change will make the text easier to follow. 

2. As validation of the VR Realism Scale in a different context (i.e., Polish) from where it was developed, is one of the main goals of this paper, measurement invariance across cultures is the key point. What measures did you take to ensure the same meaning of the obtained Polish version scale as that of the German version scale? Who were responsible for back-translating, or comparing the two versions? What were their qualifications for doing so? How did you decide the final version? Could you please provide more details about the scale translation process? 

Thank you for paying attention to the missing details, we are pleased to complete the text with all the information requested by the Reviewer. In particular, we added more information about translation process, and we tested measurement invariance. The authors of original scale published only exploratory factor analysis and they did not provide any extra variables; therefore, we could not compare results regarding relationships with other variables. 

3. A related question, what changes did you make to the original scale? You introduced that “examine the psychometric properties of the scale in the context of state-of-the-art VR technology and modern IVEs, as CAVES and modern headsets differ greatly in terms of the characteristics of the stimuli they deliver (e.g., Mestre, 2017)”. As the technological environments focused are different, what changes did you do for the original one to fit the situation of VR headsets you were specific on? 

Thank you for that remark. To clarify, we did not make any changes to the original scale. We chose the VR Realism Scale for the purpose of this validation, as we believe that its items are universal and thus can be tested with different hardware and technology (and we see it as a big advantage of this scale). The items of the scale do not contain any terms or expressions specific to a certain technology. 

Our aim was to test if this scale works well with modern technology and we believe that our results prove that it does in fact work well, without making any changes. 

We have added a sentence explaining this reasoning in the text as well.

4. I am not clear about the implication of the bi-factor model result. Is it possible to support the existence of the method effect, as all items were reported by the participates self? And if you could justify that bi-factor model support the one combined structure of simulation Realism, how should we measure this construct, as one general dimension or as multi-dimensions? A related question, why did you treat it as both unidimensional and multi-dimensional in the following analyses?

A bifactor model is a structure where all items load on a general factor but they also load on orthogonal grouping factors. The general factor represent what is common among the items whereas grouping factors (subscales) explain item response variance not accounted for by the general factor. However, we do not think that grouping factors represent a noise of measurement (ie. Method effect). However, we do not rule out that simulation realism scale produces the method effect. We suspect that correlations with similar variables that are lower than expected can be a result of the method effect.

We believe the bifactor model is handy as we can treat simulation realism as one-dimensional and multidimensional construct at the same time. In the literature, there are many VR realisms (ei. Pictorial realism, behavioral realism, interaction realism, etc.). We did not want to impose an answer to the question whether they are aspects of one simulation realism, or they are separate constructs. Thus, we used bifactor model so that researchers could choose their own preference. Since you asked, we added our recommendation for future usage of the scale to the discussion section. 

5. I agree with that it is necessary to examine the relations of simulation realism to other variables. But why did you theorize a mediation model wherein aesthetics as the mediator between simulation realism and positive emotions? Could you please provide more powerful arguments on this study, and explain why this mediation model is needed or helpful to achieve your goal of investigating its relations?

Thank you for this remark, we described our assumption about this relationship in more detail in the introduction. We add a justification of mediation analysis choice in Data analysis strategy section. And we elaborate more about consequences of this relationship in the discussion.

---

## [Decision Letter · Decision Letter 1]

24 Sep 2021

PONE-D-21-05975R1VR Realism Scale – revalidation of contemporary VR headsets on a Polish samplePLOS ONE

Dear Dr.Lipp,

Thank you for submitting your manuscript to PLOS ONE. After careful consideration, we feel that it has merit but does not fully meet PLOS ONE’s publication criteria as it currently stands. Therefore, we invite you to submit a revised version of the manuscript that addresses the points raised during the review process.

We look forward to receiving your revised manuscript.

Kind regards,

Meng-Cheng Wang

Academic Editor

PLOS ONE

Journal Requirements:

Reviewers' comments:

Reviewer's Responses to Questions

**Comments to the Author**

1. If the authors have adequately addressed your comments raised in a previous round of review and you feel that this manuscript is now acceptable for publication, you may indicate that here to bypass the “Comments to the Author” section, enter your conflict of interest statement in the “Confidential to Editor” section, and submit your "Accept" recommendation.

Reviewer #1: All comments have been addressed

Reviewer #2: All comments have been addressed

2. Is the manuscript technically sound, and do the data support the conclusions?

Reviewer #1: Yes

Reviewer #2: Yes

3. Has the statistical analysis been performed appropriately and rigorously? 

Reviewer #1: Yes

Reviewer #2: Yes

4. Have the authors made all data underlying the findings in their manuscript fully available?

Reviewer #1: Yes

Reviewer #2: No

5. Is the manuscript presented in an intelligible fashion and written in standard English?

Reviewer #1: Yes

Reviewer #2: No

6. Review Comments to the Author

Reviewer #1: Both in CFA and Bifactor model, the authors used the fixed load method (e.g., the factor loading for the first item of per factor is fixed to 1), but I’m a little unsure that the standardized factor loading of the first item is also 1, please check it carefully.

Reviewer #2: I am sorry for my delayed comments as it is hard to complete everything in time during the recent surge of pandemic.

I appreciate all your efforts to improve this paper and most of my comments have been addressed satisfactorily. There is only one minor concern: could you please reorganize your introduction session to make it more concise, it is quite long and hard o get the main purpose/ point. For example, you can briefly interpret why you consider “classical aesthetics” as the mediator in introduction session, and then add the “hypotheses session” to provide more arguments for that.

The comments above is for your reference. Good luck!

7. PLOS authors have the option to publish the peer review history of their article (what does this mean?). If published, this will include your full peer review and any attached files.

Reviewer #1: No

Reviewer #2: No

---

## [Author Response · Author response to Decision Letter 1]

18 Oct 2021

Journal Requirements: 

We reviewed our reference list – there are no retracted positions. However, due to changes made to the text, the order of position has been changed. 

Reviewer 1 

1. Both in CFA and Bifactor model, the authors used the fixed load method (e.g., the factor loading for the first item of per factor is fixed to 1), but I'm a little unsure that the standardized factor loading of the first item is also 1, please check it carefully. 

Thank you for your vigilance. You were right from the beginning – standardized factor loadings are not fixed to 1, so we've corrected the figures. 

Reviewer 2 

1. There is only one minor concern: could you please reorganize your introduction session to make it more concise, it is quite long and hard o get the main purpose/ point. For example, you can briefly interpret why you consider "classical aesthetics" as the mediator in introduction session, and then add the "hypotheses session" to provide more arguments for that. 

Thank you for this remark. We followed your suggestion and provided arguments about aesthetics in the 'Aims of our study' section next to other hypotheses. 

In addition to your comment about reorganizing the introduction, you pointed out that we don't share data and the language errors in the text. We have completed the dataset with the data (to calculate measurement invariance) which we received from the German team. These data can be found in the spreadsheet named 'CFA+item_analysis_invariance.' Additionally, a professional English language editor proofread the text, correcting all errors.

---

## [Editor Report · Decision Letter 2]

6 Dec 2021

VR Realism Scale – revalidation of contemporary VR headsets on a Polish sample

PONE-D-21-05975R2

Dear Dr. Lipp,

We’re pleased to inform you that your manuscript has been judged scientifically suitable for publication and will be formally accepted for publication once it meets all outstanding technical requirements.

Kind regards,

Meng-Cheng Wang

Academic Editor

PLOS ONE
---

## [Editor Report · Acceptance letter]

10 Dec 2021

PONE-D-21-05975R2 

VR Realism Scale – revalidation of contemporary VR headsets on a Polish sample 

Dear Dr. Lipp:

I'm pleased to inform you that your manuscript has been deemed suitable for publication in PLOS ONE. Congratulations! Your manuscript is now with our production department. 

Kind regards, 

on behalf of

Dr. Meng-Cheng Wang 

Academic Editor

PLOS ONE